# The Successful Recovery of a Critically Ill COVID-19 Patient, Following the Combination of Therapeutic Plasma Exchange and Convalescent Plasma Transfusion: A Case Report

**DOI:** 10.3390/medicina58081088

**Published:** 2022-08-12

**Authors:** Alexandru Noris Novacescu, Georgiana Duma, Dorel Sandesc, Teodora Sorescu, Monica Licker

**Affiliations:** 1Doctoral School, ‘Victor Babes’ University of Medicine and Pharmacy, 300041 Timisoara, Romania; 2Anesthesia and Intensive Care Unit, ‘Dr. Victor Babes’ Infectious Diseases and Pneumology Clinical Hospital, 300310 Timisoara, Romania; 3Anesthesia and Intensive Care Unit, ‘Dr. Teodor Andrei’ Municipal Hospital, 305500 Lugoj, Romania; 4Department of Anaesthesia and Intensive Care, ‘Victor Babes’ University of Medicine and Pharmacy, 300041 Timisoara, Romania; 5Anesthesia and Intensive Care Unit, ‘Pius Brînzeu’ Emergency Clinical County Hospital, 300723 Timisoara, Romania; 6Department of Diabetes, Nutrition and Metabolic Diseases, ‘Victor Babes’ University of Medicine and Pharmacy, 300041 Timisoara, Romania; 7Department of Diabetes, Nutrition and Metabolic Diseases, ‘Pius Brînzeu’ Emergency Clinical County Hospital, 300723 Timisoara, Romania; 8Department of Microbiology, ‘Victor Babes’ University of Medicine and Pharmacy, 300041 Timisoara, Romania; 9Multidisciplinary Research Center on Antimicrobial Resistance, ‘Victor Babes’ University of Medicine and Pharmacy, 300041 Timisoara, Romania; 10Bacteriology Laboratory, ‘Pius Brînzeu’ Emergency Clinical County Hospital, 300723 Timisoara, Romania

**Keywords:** COVID-19, therapeutic plasma exchange, convalescent plasma transfusion, case report

## Abstract

With an intricate symptom pattern involving a dysregulated host response to infection, the severe acute respiratory syndrome coronavirus 2 (SARS-CoV-2) can cause severe inflammation and cytokine storms, acute respiratory distress syndrome, coagulopathy, multi-organ failure, and finally death. The uniqueness of this case report lies in the nature of the therapeutic intervention performed. While numerous studies are available on both the use of therapeutic plasma exchange in coronavirus disease 2019 (COVID-19) patients and convalescent plasma transfusion as separate treatment methods, there is very little information regarding the combination of these procedures. We present the case of a 52-year-old male, unvaccinated for COVID-19, who tested positive on reverse transcriptase polymerase chain reaction for SARS-CoV-2 for the first time and presented in the emergency room with fever, chills, severe cough, tachypnea, tachycardia, and dyspnea that started two days before presentation. Upon rapid assessment, the patient showed signs of acute respiratory failure, so it was decided to transfer the patient to the intensive care unit, COVID-19 ward, after preliminary radiological examination. For the next 24 days, the patient was stationed in the intensive care unit, where he was closely monitored and treated. Invasive mechanical ventilation was required following the initial worsening of his respiratory status. We performed therapeutic plasma exchange on the first day of his stay in the intensive care unit, and immediately after the procedure, the patient was transfused with 500 mL of convalescent plasma from healthy donors. The patient’s condition improved over the next few days, which led to the cessation of mechanical ventilation and, after treating the superinfection, the patient was discharged home, making a full recovery. The early initiation of therapeutic plasma exchange followed by transfusion of convalescent plasma in severe and critical forms of COVID-19 may reduce the risk of the progression of the disease and ultimately reduce the risk of negative outcomes in a selected group of patients.

## 1. Introduction

The end of 2019 was marked by the emergence of a pandemic that brought an intricate symptom pattern, ranging from acute lung injury and acute respiratory distress syndrome (ARDS), systemic hyperinflammation, cytokine release syndrome, and coagulopathy to disseminated intravascular coagulopathy and thrombosis, all under the same name: COVID-19 [1].

While numerous treatment options have been investigated, some studies [2,3,4,5] have shown that the transfusion of convalescent plasma (CP) might stop the progression of the disease, while others [6,7,8,9,10,11] have suggested that therapeutic plasma exchange (TPE) might play a role in regulating the systemic hyperinflammation and coagulopathy. Even fewer studies have investigated the combination of the two procedures: transfusion of CP after the end of the TPE session [12].

## 2. Case Report

A 52-year-old male, unvaccinated for COVID-19, presented on January 2021 in the emergency room (ER) of ‘Dr. Teodor Andrei’ Municipal Hospital (Lugoj, Romania), a tertiary care hospital, an adjunct of ‘Pius Brînzeu’ Emergency Clinical County Hospital (Timisoara, Romania), with malaise, severe cough, and dyspnea, lasting for 2 days, and tested positive for COVID-19 on reverse transcriptase polymerase chain reaction (RT-PCR) 2 days before presentation—this being his first COVID-19 infection. The patient had the following comorbidities: arterial hypertension that was under treatment and obesity (BMI = 33.2 kg/m^2^), with no significant past interventions.

Assessment in the ER revealed fever (38.2 °C), tachypnea (40 breaths/min), tachycardia (110 beats/min), and signs of acute respiratory failure (SpO_2_ = 60%); therefore, the patient was transferred to the intensive care unit (ICU). Written informed consent for publication of this case report and any accompanying images was obtained from the patient in December 2021.

Timeline
PresentationA 52-year-old man presented in the ER with malaise, fever, severe cough, tachypnea, tachycardia, and dyspnea, which started 2 days before the presentation. Upon rapid assessment, the patient had low oxygen saturation and showed signs of respiratory failure. The decision was made to transfer the patient to the ICU after preliminary radiological examination.InitialtreatmentUpon ICU admission, the patient was immediately started on high flow nasal oxygenation (60 L/min, FiO_2_ = 100%) combined with a non-rebreathing oxygen mask (oxygen flow rate 15 L/min). Antiviral therapy with remdesivir, high dose corticosteroid pulse therapy with methylprednisolone, and therapeutic anticoagulation with nadroparine were started, according to hospital and national guildelines.Day 1Patient’s respiratory status worsened, requiring escalation to non-invasive ventilation with CPAP facemask and, 3 h later, after rapid assessment, intubation and mechanical ventilation were considered vital.A dual lumen 14 French dialysis catheter was placed in the right femoral vein under echographic guidance and a single TPE session was performed using 40 mL/kg FFP as substitute. Upon completion of the TPE session, the patient was transfused with 500 mL ABO compatible CP under careful monitorization. (The procedures were performed on day 3 after the onset of symptoms).Day 9Sedation and neuromuscular blockade were ceased and respiratory weaning from mechanical ventilation was started by switching the ventilation mode to spontaneous (CPAP).Day 11Extubation was performed.Day 12Uncontrollable hypercapnia leading to neurologic status aggravation determined intubation and the start of mechanical ventilation and continuous sedation.Day 15Patient was weaned off mechanical ventilation and extubated. RT-PCR test resulted negative and the patient was transferred to the non-COVID-19 ICU.Day 23Lung CT was performed for pulmonary re-evaluation.Day 24The patient was discharged home.

A computerized tomography (CT) scan was unavailable at that time, so preliminary radiologic examination consisted of a posteroanterior chest X-ray (Figure 1). 

Upon ICU admission, the patient had an SpO_2_ of 65% on a non-rebreather oxygen mask, with an oxygen flow rate of 15 L/min. He was immediately started on high-flow nasal oxygen (oxygen flow rate 60 L/min, fractional inspired oxygen (FiO_2_) 100%) combined with a non-rebreather oxygen mask (oxygen flow rate 15 L/min), resulting in an SpO_2_ of 88%. A urinary catheter was set in place and an arterial line was inserted into the left radial artery to draw blood for arterial blood gases and routine laboratory investigations. After 1 h of high-flow oxygenation, the arterial blood gas (ABG) analysis revealed a P_a_O_2_ of 54 mmHg, and combined with the chest X-ray results, the diagnosis of ARDS was established. ARDS was defined as acute-onset hypoxemia (the partial pressure of arterial oxygen to FiO_2_ ratio (P/F ratio) < 300), with >50% bilateral pulmonary opacities on chest imaging within 24–48 h that were not fully explained by congestive heart failure and that required ICU treatment and monitoring [13]. By correlating the clinical and radiological findings, the patient’s form of disease was considered critical, consistent with the COVID-19 guidelines [14,15], with a reserved prognosis.

Antiviral therapy was initiated with remdesivir (Veklury; Gilead Sciences, Carrigtwohill, Ireland), at a loading dose of 200 mg, followed by 100 mg o.d. for 5 days. High-dose corticosteroid pulse therapy with methylprednisolone (Lemod Solu; Hemofarm, Timisoara, Romania), 250 mg b.i.d. for 3 days was also started, and anticoagulation consisted of therapeutic dose of nadroparine (Fraxiparine; Aspen Pharma Trading, Dublin, Ireland), once every 12 h, according to body weight (5700 Anti-Xa IU), all according to hospital and national guidelines.

The following morning, the patient’s respiratory status worsened, requiring escalation to continuous positive airway pressure (CPAP) noninvasive ventilation with a FiO_2_ of 100%, resulting in an SpO_2_ of 93%. Three hours later, the patient became tachycardic (130 beats/min) and tachypneic (43 breaths/min), SpO_2_ decreased to 68%, and his mental status altered (HACOR score 24). Urgent orotracheal intubation was decided (the patient presented difficult intubation, with a Cormack–Lehane score of 3) and mechanical ventilation was initiated (BiLevel ventilation mode, with a FiO_2_ of 100%, an inspiratory pressure of 24 cmH_2_O, respiratory frequency of 20 breaths/min, and positive end-expiratory pressure (PEEP) of 10 cmH_2_O), resulting in a driving pressure of 14 cmH_2_O. The patient was started on continuous sedation and neuromuscular blockade after inserting a central line in the right internal jugular, under echographic guidance. Two hours after the initiation of mechanical ventilation, ABG analysis showed a P_a_O_2_ of 87 mmHg, SO_2_ of 94%, P_a_CO_2_ of 74 mmHg, pH of 7.165, and an anion gap of 19 mmol/L, specific for decompensated respiratory acidosis. The P/F ratio was 87. In a rescue attempt to stop the specific evolution of the COVID-19 pneumonia, we placed a dual lumen, 14 French, dialysis catheter in the right femoral vein under echographic guidance and performed a single TPE session on the INFOMED HF 440 machine, with a plasma/blood separation of 20%, using 40 mL/kg fresh frozen plasma as a substitute. Heparin, 500 U/mL, was used for the continuous anticoagulation of the circuit. Upon completion of the TPE session, the patient was transfused with 500 mL ABO compatible (A II Rh-positive) CP under careful monitoring. An amount of 500 mL CP was used because of the low neutralizing antibody titer ratio [1 bag of 250 mL, titer 1:80 and 1 bag of 250 mL, titer 1:160, values considered acceptable, if higher titer units were not available (https://www.uptodate.com/contents/COVID-19-convalescent-plasma-and-hyperimmune-globulin) (accessed on 10 May 2022)]. No adverse reactions were noted.

Over the next few days, we were able to decrease the use of mechanical ventilation, reaching an inspiratory pressure of 22 cmH_2_O, a PEEP of 5 cmH_2_O, a respiratory frequency of 16 breaths/min, and FiO_2_ of 45%. ABG revealed a pH of 7.38, P_a_CO_2_ of 60 mmHg, P_a_O_2_ of 75 mmHg, sO_2_ of 94%, and an anion gap of 9 mmol/L. The prone position was considered at the beginning, but because in the following days we were able to constantly reduce the FiO_2_ and the mechanical ventilation support, it was no longer necessary. An improvement in oxygenation and a decrease in inflammatory markers was noticed (Figure 2).

On day 9, mucopurulent discharge was noticed on the tracheal tube along with a spike in temperature of 38.5 °C. The orotracheal tube was changed, culture from bronchial lavage was obtained, blood culture and urine culture were also obtained and the patient was started on broad spectrum antibiotics, consistent with the local ICU flora [meropenem (Meropenem Kabi; Fresenius Kabi, Bad Homburg, Germany), 1 g every 8 h, vancomycin (Vancomycin Kabi; Fresenius Kabi, Bad Homburg, Germany), 1 g every 12 h, and polymyxin E (Colistin Atb.; Antibiotice a+, Iasi, Romania), 3 million IU every 8 h, following a loading dose of 9 million IU]. Neuromuscular blockade was stopped as was sedation, and after showing signs of spontaneous breathing, the patient was switched to a spontaneous breathing mode (CPAP, FiO_2_ of 40% with a pressure support of 10 cmH_2_O).

On day 10, the urine culture results came back negative.

On day 11, the patient’s mental and respiratory status allowed extubation after a successful spontaneous breathing trial. Oxygen was delivered on a non-rebreather oxygen mask with a flow of 10 L/min, resulting in a SpO_2_ of 95%; ABG analysis revealed: pH 7.4, P_a_CO_2_ 56 mmHg, P_a_O_2_ 91 mmHg, and anion gap 7.6 mmol/L.

On the morning of day 12, the patient’s condition worsened, a depressed level of consciousness was noticed (Glasgow Coma Scale of 6), and ABG showed a P_a_CO_2_ of 130 mmHg. We decided to quickly intubate the patient and to resume mechanical ventilation and continuous sedation.

On day 13, the results returned from the bronchoalveolar lavage culture were positive for *Acinetobacter baumannii* and *Candida* sp. Antimicrobial susceptibility testing revealed an extensively drug resistant *A. baumannii* strain (XDR), with resistance to all but two categories of antimicrobials (piperacillin/tazobactam and colistin). Antibiotics were adjusted accordingly [meropenem was de-escalated to piperacilin/tazobactam (Piperacilin/Tazobactam Kabi 4 g/0.5 g; Fresenius Kabi, Bad Homburg, Germany) 4.5 g every 6 h], and an antifungal was added [fluconazole (Fluconazole; B Braun, Hessen Germany), at a loading dose of 400 mg, followed by 200 mg o.d.)]. Aerosols with polymyxin E (Colistin Atb.; Antibiotice a+, Iasi, Romania) (1 million IU) were also administered, q.i.d., at 6 h intervals.

On day 14, the blood culture returned a positive result for the same pathogens: *A. baumannii* (with the same pattern of resistance) and *Candida* sp. Given the fact that no gram-positive bacteria could be identified from the cultures, vancomycin was stopped. 

On day 15, after a progressive improvement in the patient’s respiratory status, sedation was stopped, and the patient was extubated after a successful spontaneous breathing trial. RT-PCR for COVID-19 from broncheoalveolar lavage was performed on the same morning, with a negative result. The patient was moved to the non-COVID-19 ICU.

Over the next few days, the patient gradually improved, physiotherapy was started, and on day 23, a lung CT was performed to reevaluate the extent of the pulmonary damage (Figure 3). Sputum culture and blood cultures were also obtained.

On day 24, the culture results returned negative, and the patient was discharged.

Three months after being discharged, the patient underwent a follow-up chest CT, showing full resolution of the lung damage (Figure 4) and made a full recovery.

## 3. Discussion

After reviewing the available literature, Liu et al. [4] found that CP transfused even two weeks after the onset of manifestation could improve the symptoms and mortality rate in patients with severe or critical forms of COVID-19.

Balagholi et al. [10] stated that the main factor in the success of TPE is starting the procedure in the early stage of inflammation when there is a high concentration of inflammatory cytokines and abnormal coagulation agents. The duration from symptom onset to the initiation of TPE and CP transfusion in the present case was 3 days.

Honore et al. [16] found that specific SARS-CoV-2 IgG and IgA antibodies can be detected in the waste bag plasma, and that at the same time the circulating number of antibodies is reduced. TPE relieves the burden of pro-inflammatory markers (e.g., cytokines, chemokines, PAMPs, DAMPs), pro-permeability factors (e.g., angpt-2, Hpa-1), and pro-coagulation factors (e.g., vWF-M), while also replacing consumed protective factors that are critical to maintaining microcirculatory flow (e.g., ADAMTS-13, AT-III, and protein C) and preventing vascular leaks (e.g., angiopoietin-1, sTie-2, Hpa-2), while the transfusion of CP helps in boosting the number of SARS-CoV-2 specific antibodies and in immune-restoration (IgG, IgM, IgA).

In a small, non-randomized, case-controlled clinical trial, investigating the potential benefits of the sequential use of TPE followed by the transfusion of CP performed early, Novacescu et al. [17] found a statistically significant difference in mortality, oxygenation, and inflammatory markers between the intervention and control groups.

The patient’s improved outcomes concerning survival, oxygenation, and inflammation are in line with the results of the aforementioned studies. However, it is difficult to establish whether the combined effects of these two therapies show superiority over the single use of each one of them. The use of corticosteroids and antiviral agents may also have played a role in the outcomes.

Despite the fact that on days 13 and 14 of hospitalization, a bacterial and fungal superinfection was found, the evolution of the patient was favourable following the treatment administered. There is data in the literature regarding the use of nebulized colistin which may improve the outcomes of pulmonary infections caused by both *A. baumanni* and SARS-CoV-2 [18].

Dysregulated inflammation syndrome in severe forms of COVID-19 requiring ICU therapy may be controlled by the early initiation of TPE followed by the transfusion of CP in order to relieve inflammation, prevent cytokine storms, improve oxygenation, and eliminate viral load and autoantibodies by shifting the antigen-antibody ratio in favour of the latter [1,19].

A major advantage of this case report is that it presents a rescue treatment option that has been explored very little, which could have a huge impact on critically ill patients suffering from severe forms of COVID-19 and could possibly prove to be useful in other diseases as well. The positive outcome regarding mortality and inflammatory markers is also a strength of this study. There are some limitations associated with this case report: chest CT was unavailable for initial evaluation upon admission to the ER; IL-6 testing was unavailable in our laboratory; antivirals, corticosteroids, and mechanical ventilation may have interfered with the discussed outcomes; sequencing of the SARS-CoV-2 variant was not performed, neither from the patient nor from the convalescent plasma; and the patient’s baseline anti-SARS-CoV-2 antibody titer was not determined. The fact that high-titer convalescent plasma was not available for transfusing the patient is also a limitation of this report.

The present case report showed that the early use of TPE followed by transfusion of CP resulted in a reduction of inflammatory markers, improved oxygenation, and overall in an improved outcome for our patient. The extent to which antivirals and corticosteroids interfered with the outcome is difficult to assess. More research should be conducted, including large randomized controlled trials, to further explore this innovative treatment and determine the characteristics of patients who may potentially benefit the most from this treatment combination.

## Figures and Tables

**Figure 1 medicina-58-01088-f001:**
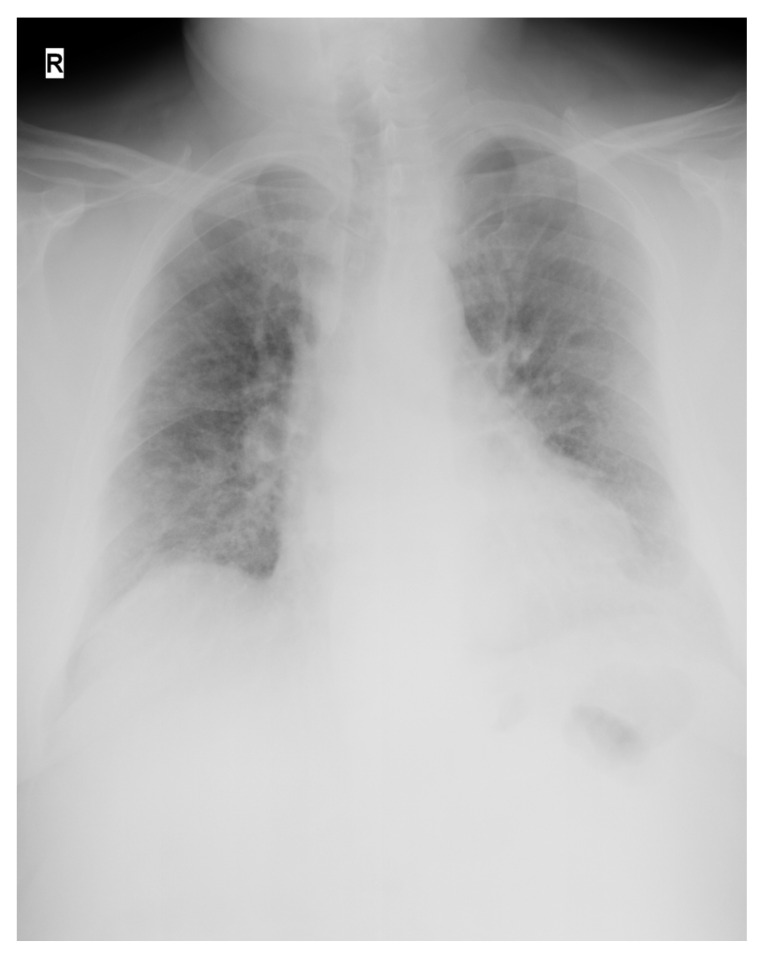
Posteroanterior chest X-ray at presentation in the ER. Multiple areas showing ground-glass opacities, on both pulmonary areas, amounting to 50% pulmonary damage. Day of admission.

**Figure 2 medicina-58-01088-f002:**
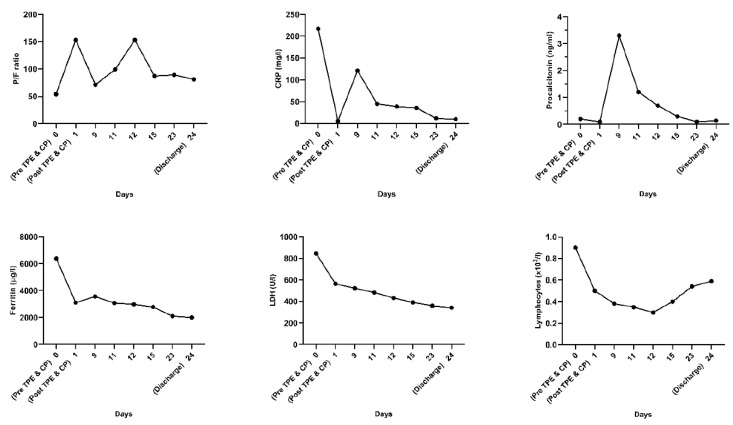
Dynamic graphic analysis of oxygenation (P/F ratio) and inflammatory markers. After the procedure (therapeutic plasma exchange followed by convalescent plasma transfusion), an improvement in oxygenation and a decrease in inflammatory markers could be noticed.

**Figure 3 medicina-58-01088-f003:**
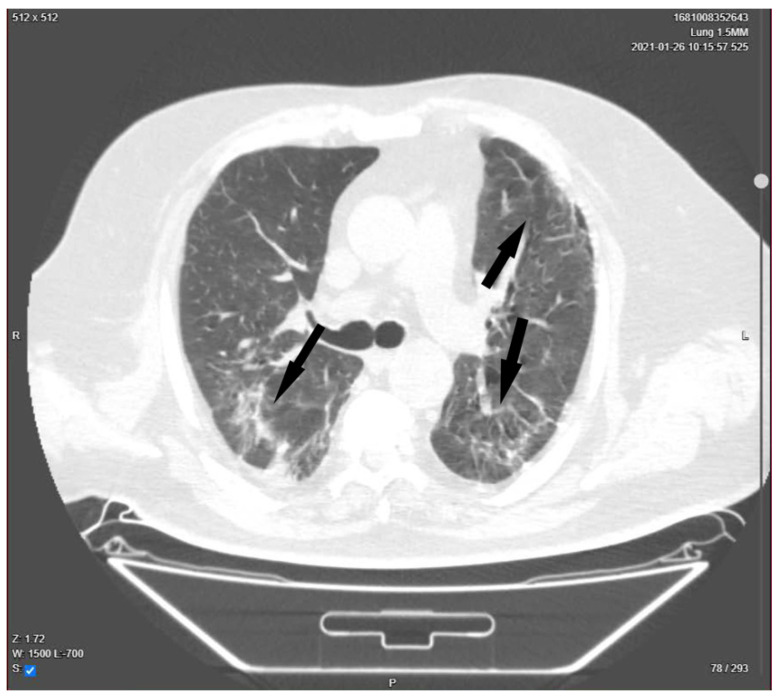
Chest CT. The arrows indicate areas of ground-glass opacities, summing up to 40% pulmonary damage. Day 23 of admission.

**Figure 4 medicina-58-01088-f004:**
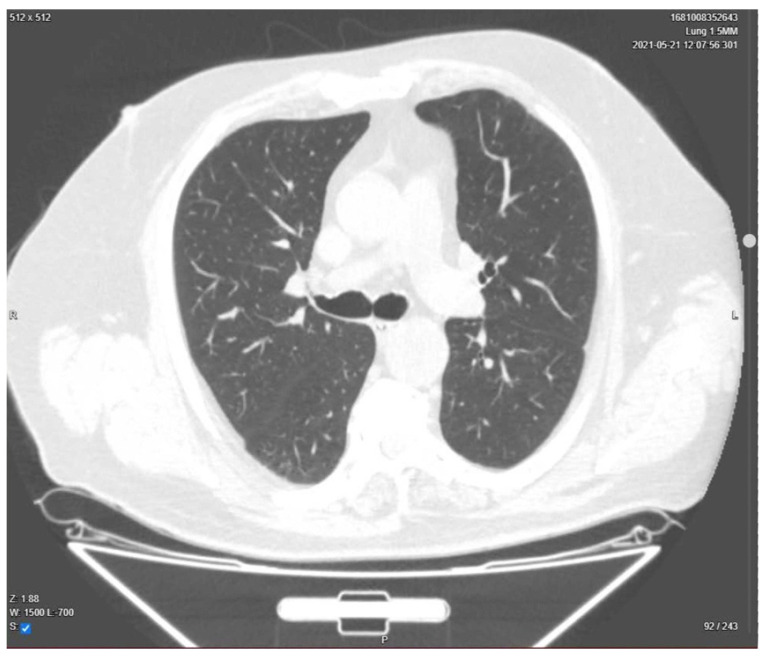
Chest CT. Complete resolution of the pulmonary damage. Day 90 from discharge.

## Data Availability

Data is contained within the article.

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
