# Peer review of "The Successful Recovery of a Critically Ill COVID-19 Patient, Following the Combination of Therapeutic Plasma Exchange and Convalescent Plasma Transfusion: A Case Report"

_medicina, 2022, doi:10.3390/medicina58081088_

Round 1

Reviewer 1 Report

I read with great interest a case report on the sequential use of therapeutic plasma exchange (TPE) and convalescent plasma in a patient with Critical COVID-19.

I want to congratulate the authors on the wonderful work done in managing this critically ill patient.

I have a few suggestions to improve the readability of this case report.

1. Avoid using the acronym CVP for convalescent plasma as it creates confusion with the acronym central venous pressure. 

2. Serial inflammatory marker (IL-6, CRP, Procalcitonin, LDH, Ferritin, etc.) trend, if available (not only on day one and day 7), would have been helpful. The values pre-and post- TPE are also useful to understand the role of TPE in reducing cytokines.

3. Consider removing the phenotypes L and H in the case discussion. As they do not have any clinical implications for this case and divert the attention of the readers.

4. Finally, use arrows in the CT images to show the areas of pathology.

Reviewer 2 Report

Novacescu et al. report the case of a 52 y/o patient with Covid-19 who was treated with early PEx and convalescent plasma. Although the case is of interest, I have a couple of concerns and questions.

- What was the patient's baseline anti-SARS-CoV-2 antibody titer? Was this his first Covid-19 infection?

- Which SARS-CoV-2 variant did the patient have? Was sequencing performed?

- What SARS-CoV-2 variant was used in the convalescent plasma? Did the two match?

- When was the treatment with corticosteroids initiated? Guidelines recommend the use of steroids only from day 5 onwards.

- Why was the patient put on therapeutic anticoagulation? There is little evidence to support higher-dose anticoagulation in patients with Covid-19.

- IL-6 levels would also be interesting to assess treatment response

- There is little evidence to support the routine use of convalescent plasma in patients with Covid-19. The putative benefit of convalescent plasma is only suggested if the patient is not immunocompetent and when used early in the course of the disease.

- The combination of the two treatments is interesting, however, there are  significant disadvantages thereof: costly treatment (PEx and convalescent plasma), not available everywhere, plasma of convalescent patients may not target newer variants. Only a selected group of patients may potentially benefit from this treatment combination - of which there is currently no evidence. However, I am glad that the treatment worked for this patient and he had a full recovery.

- Also, I am curious about the timing of the two treatments. Convalescent plasma should be given as early as possible, when there are no endogenous antibodies yet. However, peak inflammation levels can take time and may present approximately around day 5-10. What do the authors suggest, if both treatments are used? Is there literature available for these instances?

- Some information is not pertinent the presentation of this case and can be deleted from the text.

Round 2

Reviewer 2 Report

The authors answered most of my questions. However, I feel that some of it should be included in the manuscript (perhaps as "limitations").
